# Impact of Elevated LDH on Cystatin C-Based Glomerular Filtration Rate Estimates in Patients with Cancer

**DOI:** 10.3390/jcm11185458

**Published:** 2022-09-16

**Authors:** Enver Aydilek, Manuel Wallbach, Michael Koziolek, Gerald Georg Wulf, Nils Brökers

**Affiliations:** 1Department of Hematology and Medical Oncology, University Medicine Göttingen, 37075 Göttingen, Germany; 2Department of Nephrology and Rheumatology, University Medical Center Göttingen, 37075 Göttingen, Germany; 3German Centre for Cardiovascular Research (DZHK), Partner Site Göttingen, 37075 Göttingen, Germany

**Keywords:** cystatin c, ldh, eGFR, cystatin C-based glomerular filtration, cancer, oncology, hematology

## Abstract

**Background:** The determination of renal function is crucial for the clinical management of patients with cancer. The glomerular filtration rate (GFR) serves as a key parameter, estimated by creatinine clearance determination in 24-h collected urine (CrCl) as well as equation-based approaches (eGFR) relying on serum creatinine (eGFR CKD EPIcrea) or serum cystatin C (eGFR cystatin C). Serum creatinine and serum cystatin C levels differentially depend on muscle and tumor mass, respectively. Although muscle and tumor mass may thus represent confounding factors, comparative studies for eGFR estimate approaches in cancer patients are lacking. **Methods:** The present study retrospectively analyzed GFR estimates based on equations of creatinine (eGFRcr), cystatin C (eGFRcys) and combined creatinine-cystatin C levels (eGFRcr-cys) in a subset of patients. The associations of LDH with cystatin C or LDH with eGFRcr, eGFRcys and GFRcr-cys were explored. **Results:** The laboratory values of 123 consecutive patients were included. The median age was 59 (24–87) and 47.2% were female. There was a statistically significant difference in the mean of CKD EPIcrea (85.17 ± 21.63 mL/min/1.73 m^2^), CKD EPIcys (61.16 ± 26.03 mL/min/1.73 m^2^) and CKD EPIcrea-cys (70.42 ± 23.89 mL/min/1.73 m^2^) (*p* < 0.0001). Spearman’s correlation analysis revealed a significant correlation of elevated plasma LDH >1.5 UNV and cystatin C values (r = 0.270, *p* < 0.01, n = 123). LDH values >1.5 UNV were associated with significantly lower CKD EPIcys (r = 0.184, *p* < 0.01) or CKD EPIcrea-cys (r = 0.226, *p* < 0.05) estimates compared to CKD EPIcrea. **Conclusions:** The inclusion of cystatin C as a biomarker led to a lower eGFR estimates compared to creatinine alone or in a combination of both cystatin C and creatinine. The level of cystatin C correlated with the level of LDH, suggesting that the use of cystatin C-based calculations of GFR in cancer patients with elevated LDH should be used with caution.

## 1. Introduction

The accurate determination of renal function is essential in the routine clinical care of cancer patients [1]. Underestimation or overestimation can lead to the over- or under-dosing of chemotherapeutic agents, inappropriate drug selection leading to therapeutic failure or the occurrence of increased treatment-related toxicity [2]. 

The exact determination of renal function by inulin or radioisotope is cost-intensive, time-consuming and requires intervention. The determination of the glomerular filtration rate (GFR) by using 24 h collection (CrCL) is error-prone and requires adequate patient compliance. Equations for the estimation of GFR based on creatinine, cystatin C or both have been developed in recent decades [3,4]. However, serum creatinine levels may be influenced by age and muscle mass, especially in cancer patients, due to tumor-associated sarcopenia [5].

Cystatin C is available as an additional biomarker for estimating GFR [6], and previous studies revealed substantial intraindividual differences in GFR estimates using the cystatin C-based equation (eGFRcys) and the creatinine-based GFR equation (eGFRcr) [7,8]. The interpretation and clinical impact of differing eGFRs have not been clarified. The widely applied creatinine-based GFR (CKD EPIcrea) integrates age and sex but does not consider muscle mass, diet or physical activity, which may result in the over- or underestimation of renal function [9,10]. The National Kidney Foundation (NKF) and the American Society of Nephrology (ASN) have recommended the increased use of cystatin C and the combined equation with creatinine and cystatin C to estimate the glomerular filtration rate [11].

The cysteine protease inhibitor cystatin C is synthesized in all nucleated cells [12,13,14], including cancer cells. It is catabolized completely in the proximal renal tubule and is not returned to circulation after glomerular filtration. For this reason, it is an ideal marker for estimating the glomerular filtration rate [15]. However, in cancer patients, a high tumor cell burden may lead to an increased release of cystatin C into the circulation [16,17,18]. LDH represents an established parameter to estimate tumor cell mass and is part of various prognosis scores [19]. The effect of steroids and additional diseases like diabetes or systemic inflammation on cystatin C expression is also under controversy [20,21].

To our knowledge, no comparative studies have been performed on cancer patients in which tumor mass was also added as a confounding variable. In this study, we examined the results obtained with the different equations and the influence of tumor mass and discuss possible implications for clinical practice. 

## 2. Methods

Serums were collected from all patients as part of routine diagnostics at the time of admission and analyzed in the laboratory of the University Medicine Göttingen. The parameters (Plasma creatinine, urine creatinine, LDH and cystatin C) were measured on the Architect c16000 device from Abbott (Abbott, Chicago, IL, USA). The study was approved by the ethics committees of the University Medical Center Göttingen (*No: 19/1/22*).

### 2.1. Serum Creatinine and 24 h Collection Urine and Equation of Estimated GFR

The measurement of serum creatinine was performed by Architect c16000 Abbott in the laboratory of the University Medical Center Göttingen. For eGFR calculation, the equation of creatinine-based (eGFRcr; 2009), the equation of cystatin-C-based (eGFRcys; 2012) and the equation of combined creatinine-cystatin C-based (eGFRcr-cys; 2012) were used [22]. 

### 2.2. Statistics

GraphPad PRISM Version 9.1 (GraphPad Software, San Diego, CA, USA) and Statistical Package for the Social Sciences (SPSS) 27.0 (IBM Corp., Armonk, NY, USA) were used for the statistical analysis. For correlations, the two-tailed nonparametric Spearman test, two-tailed paired *t*-test and simple linear regression analysis were used. A one-way ANOVA Test was performed to analyze the three groups. All *p*-values are two-sided, the significance level is <0.05, and confidence intervals refer to 95% limits. For data documentation, Excel Version 2019 (Microsoft Software, Redmond, Washington, DC, USA) was used.

## 3. Results

### 3.1. Characteristics of the Study Participants

Biomarkers (creatinine, cystatin C and LDH) were retrospectively investigated from a total of 123 patients with an underlying hematological and oncological disease who were treated at the University Medical Center Göttingen. The mean age of the participants in this study was 59.39 ± 11.38 in a range of 24–87 years. The sex was equally distributed (male 52.8%, female 47.2%). The majority of the study participants had an NHL (Non-Hodgkin lymphoma) as their underlying disease (68.3%). In total, 77.3% of patients had a BMI in the range of healthy (18.5–24.9 kg/m^2^) to overweight (25–29.9 kg/m^2^). The number of underweight was only 1.6% (Table 1).

### 3.2. Estimated GFR with the Use of the Three Equations

In total, the estimated GFR was determined in 123 patients based on the measured creatinine, cystatin C or both, using equations as described by Inker et al. [21]. The mean creatinine-based equation was 85.17 ± 21.63 mL/min/1.73 m^2^ and ranged between 31 and 129 mL/min/1.73 m^2^. The mean cystatin C-based equation was 61.16 ± 26.03 mL/min/1.73 m^2^ and ranged between 15–130 mL/min/1.73 m^2^. In the combined creatinine-cystatin C-based equation, the mean was 70.42 ± 23.89 mL/min/1.73 m^2^ and ranged between 23–124 mL/mL/1.73 m^2^ (Figure 1). The one-way ANOVA test shows a significant difference between all three equations (*p* value < 0.0001, r = 0.147).

Visualized in the Bland–Altman-Plot, the difference in the mean values was 14.76 + 12.83 mL/min/1.73 m^2^ between eGFRcr and eGFRcr-cys (Figure 2). 

Depending on the formula used, there is a different allocation of stages according to KDIGO. The difference is not significant in the one-way ANOVA test (Figure 3).

### 3.3. Correlation of the Cystatin C and LDH

Correlation analysis of LDH and cystatin C was performed. Spearman’s correlation analysis between LDH and cystatin C showed a significant correlation (*p*-value < 0.01, r = 0.270; n = 123) (Figure 4).

The eGFRs based on the three equations were analyzed in relation to the plasma LDH levels. LDH values below 225 U/l (female) and 250 U/I (men) are normal and values >225 U/l (female) and >250 U/I (men) are increased. The values were divided into two groups with LDH ≤ 1.5 UNV U/l (n = 89) and >1.5 UNV (n = 34). An elevation >1.5 UNV is considered relevant and has prognostic value, e.g., in germ cell tumors. No significant difference was found using the eGFRcr. However, when using the equation eGFRcys, a significant difference was found in the Two-tailed paired *t*-test between LDH ≤ 1.5 UNV and LDH >1.5 UNV U/l (*p* < 0.01). Using the equation eGFRcr-cys, there was also a significant difference (*p* < 0.05). In the analysis of the equations among themselves (eGFRcr vs. eGFRcys and eGFRcr vs. eGFRcr-cys), a significant difference was found in both groups (LDH ≤ 1.5 UNV n = 89; *p* < 0.0001) and (LDH > 1.5 UNV n = 34; *p* < 0.0001) (Figure 5).

### 3.4. Subgroup Analysis of Patients with <1.5 UNV and >1.5 UNV Normal and Elevated LDH

The subgroup analysis showed an approximately equal distribution of the underlying primary disease. Parameters such as age and BMI were comparable. The number of female patients was higher in the group with LDH > UNV (n = 21; 61.8%) compared to LDH < 1.5 UNV (n = 37; 41.6%) (Table 2).

## 4. Discussion

This is the first study investigating different formulas to estimate GFR in cancer patients, presenting three major findings: 1. There are significant differences in eGFR calculated by different formulas, and the use of CKD-EPIcys showed significant lower renal function compared to the estimation by CKD-EPIcrea. 2. The levels of LDH correlate with the levels of cystatin C and consecutively with eGFR levels based on CKDEPIcys. 3. In patients with elevated LDH levels above >1.5 UNV as a surrogate for high tumor mass or cell turn-over, respectively, eGFR calculated by cystatin C differs from eGFRcrea, whereas eGFR calculation in patients with LDH levels below this value did not differ using creatinine or cystatin. 

Considering that cystatin C is expressed in all nucleated cells, and patients with cancer have high cell turnover, the correlation of cystatin C with LDH is of special interest [22]. The levels of LDH are suggested to be elevated in many types of cancers and have been linked to tumor growth, and are therefore also an indication of high cell turnover [23,24]. In the present study, there was a significant correlation between cystatin C with LDH, suggesting an underestimation of renal function in patients with high cell turnover or tumor mass. If cell turnover and LDH levels normalize during therapy, the cystatin C-based equation would likely show higher values for eGFR compared to initial values.

By using the eGFRcys-based equation there was a significant difference in eGFR between patients with elevated LDH levels compared with patients with normal LDH levels caused by an increased level of cystatin C in patients with elevated LDH. This might contribute to the hypothesis that in the presence of increased tumor mass or high cell turnover and associated increased cystatin C, the use of the eGFRcys equation leads to an underestimation of renal function. Thus, the use of the cystatin C-based equation must be used with caution. Underestimation, in turn, is associated with underdosing of drugs, for example, and the consequent worsening of prognosis.

The non-GFR-dependent parameters, such as muscle mass, nutritional status, CRP, albumin, leukocyte count, BMI and therapy used, as well as the use of steroids, are factors that have been partially studied in the literature and significantly influence cystatin C. It should be noted that these factors have not been systematically analyzed and should be considered in further studies. Furthermore, no gold standard is used in this study as a limitation. The determination of GFR via a 24 h collection of urine was only successful in a small part of the patients and was error-prone. Therefore, no analysis can be performed. The determination of renal function in cancer patients remains difficult and each has its own difficulties. A workable gold standard that is widely used does not exist and remains to be defined. Further prospective studies are necessary, especially in patients with underlying hematological and oncological diseases. However, it could be speculated that the combination with creatinine partly compensates for this effect and that the equation with only one of these parameters may be inferior to the combination.

## 5. Conclusions

The present study shows a statistically significant and clinically meaningful difference in estimates of eGFR by using different eGFR equations in cancer patients. As cystatin C correlates with the levels of LDH, the evaluation of eGFRcys should be interpreted with caution in patients with elevated levels of LDH, since it might contribute to an underestimation of GFR. The use of the equation with cystatin C alone or in combination with creatinine should always be performed considering high cell turnover in cancer patients. There is a special need for further studies using the different eGFR equations to optimize the determination of kidney function.

## Figures and Tables

**Figure 1 jcm-11-05458-f001:**
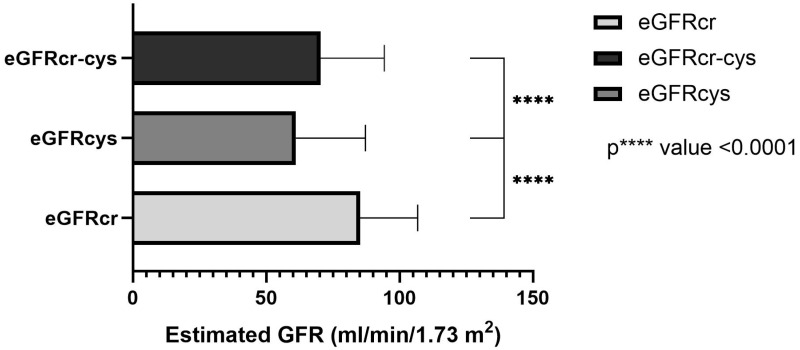
Estimated GFR (mL/min/1.73 m^2^) of the three equations which is based on creatinine (eGFRcr), cystatin C (eGFRcys) and combined creatinine-cystatin C-based (eGFRcr-cys). The one-way ANOVA test shows a significant difference between all three equations (*p* **** value < 0.0001, r = 0.147).

**Figure 2 jcm-11-05458-f002:**
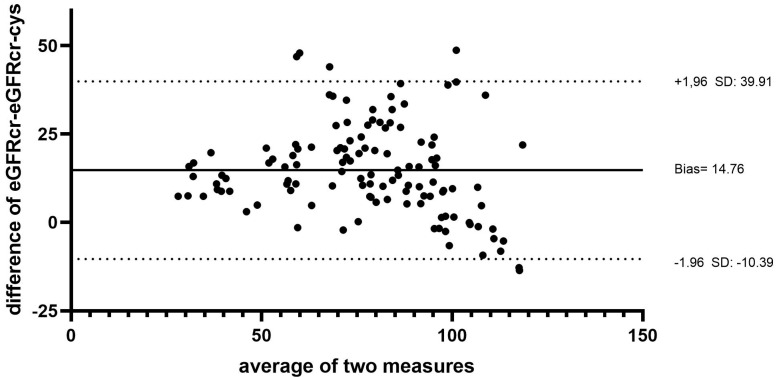
Difference between eGFRcr and eGFRcr-cys (Bland–Altman-Plot; 14.76 mL/min/1.73 m^2^). SD denotes the standard deviation of the differences between two measurements.

**Figure 3 jcm-11-05458-f003:**
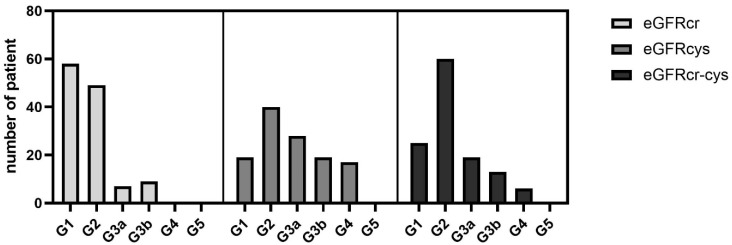
Absolute number of patients divided into the GFR category according to KDIGO. In the one-way ANOVA test, there was no significant difference among means of the three equations *p*-value = 0.391.

**Figure 4 jcm-11-05458-f004:**
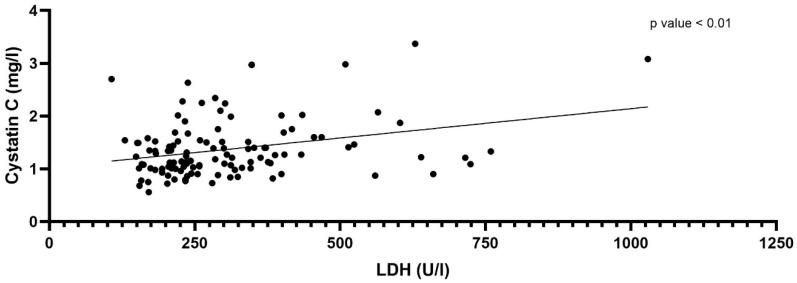
Spearman (ρ) correlation coefficient between LDH and Cystatin C (*p* < 0.01, r = 0.239, n = 120).

**Figure 5 jcm-11-05458-f005:**
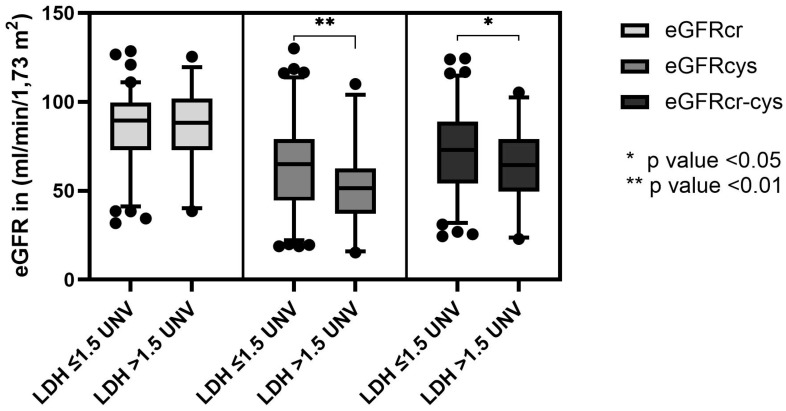
Significant difference in eGFR between LDH ≤ 1.5 UNV and >1.5 UNV in eGFRcys (*p* < 0.01) and eGFRcr-cys (*p* < 0.05) but not eGFRcr in a Two-tailed paired *t*-test.

**Table 1 jcm-11-05458-t001:** The characteristics of the included study participants.

Characteristic	
**Age**	**Years**
mean (range)	59 (24–87)
**Sex**	**n (%)**
male	65 (52.8)
female	58 (47.2)
**Disease**	**n (%)**
NHL	84 (68.3)
Solid	19 (15.4)
AML	16 (13)
MPN	2 (1.6)
Non-cancer	2 (1.6)
**BMI**	
mean (SD)	26.73 ± 5.28
	**n (%)**
<18.5 kg/m^2^	2 (1.6)
18.5–24.9 kg/m^2^	43 (35)
25–29.9 kg/m^2^	52 (42.3)
30–34.9 kg/m^2^	16 (13)
35–39.9 kg/m^2^	7 (5.7)
>40 kg/m^2^	3 (2.4)

**Table 2 jcm-11-05458-t002:** Subgroup analysis between LDH ≤ 1.5 UNV and LDH > 1.5 UNV.

Parameter	LDH ≤ 1.5 UNV	LDH > 1.5 UNV	*p*-Value
n	89	34	-
Age	61 ± 11	60 ± 12	0.82
Female n(%)	37 (41.6)	21 (61.8)	0.10
BMI	26 ± 4.6	25 ± 6.7	0.44
**Disease**	**n (%)**	**n (%)**	
NHL	59 (66.3)	25 (73.5)	-
Solid	16 (18)	3 (8.8)	-
AML	13 (14.6)	3 (8.8)	-
MPN	0	2 (5.9)	-
Non-cancer	1 (1.1%)	1 (2.9)	-
**eGFR equation**	**(mL/min/1.73 m^2^)**	**(mL/min/1.73 m^2^)**	***p*-Value**
eGFRcr	89.5 ± 21.3	88.1 ± 22.6	0.427
eGFRcys	65 ± 26.4	51.5 ± 23.3	0.004
eGFRcr-cys	73 ± 24.3	64.4 ± 21.7	0.010

## Data Availability

Not applicable.

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
