# Peer review of "Impact of Elevated LDH on Cystatin C-Based Glomerular Filtration Rate Estimates in Patients with Cancer"

_jcm, 2022, doi:10.3390/jcm11185458_

Round 1
Reviewer 1 Report
Aydilek et al. performed an interesting study on glomerular filtration rate estimates in 123 patients with cancer.
As the introduction states that « underestimation or overestimation [of GFR] can lead to over- or under-dosing of chemotherapeutic agents or inappropriate drug selection leading to therapeutic failure or occurrence of increased treatment-related toxicity », it would have been interesting to classify the patients into chronic kidney disease stages according to eGFRcr, eGFRcys and eGFRcr-cys. How many patients would have been classified among different stages between equations?
The manuscript is well-written.
A weakness would be that demonstrated significant covariates such as CRP, albumin, BMI, etc. were not collected/tested. This should be listed as a limitation. However, no gold standard has been used in this study.
Minor comments :
Line 72: Title should be written using italics.
Line 73 to 76: Which equation was used in this study? Please specify. Furthermore, there seems to be a mistake in references numeration such as reference 25 deals with steroids impact rather than GFR equation (reference 26). Please check throughout the manuscript.
Line 98 : The expression « the above equations » is confusing because no equation was written, please rephrase.
Author Response
Response to reviewer 1:
As the introduction states that « underestimation or overestimation [of GFR] can lead to over- or under-dosing of chemotherapeutic agents or inappropriate drug selection leading to therapeutic failure or occurrence of increased treatment-related toxicity », it would have been interesting to classify the patients into chronic kidney disease stages according to eGFRcr, eGFRcys and eGFRcr-cys. How many patients would have been classified among different stages between equations?
As you can see attached word file.
A weakness would be that demonstrated significant covariates such as CRP, albumin, BMI, etc. were not collected/tested. This should be listed as a limitation. However, no gold standard has been used in this study.
The following sentences are added in our discussion: It should be noted that these factors have not been systematically analyzed and should be considered in further studies. Furthermore, no goldstandard is used in this study. The determination of GFR via a 24h collection urine was only successful in a small part of the patients and was error-prone. Therefore, no analysis can be performed.
Minor comments :
Line 72: Title should be written using italics.
No problem.
Line 73 to 76: Which equation was used in this study? Please specify. Furthermore, there seems to be a mistake in references numeration such as reference 25 deals with steroids impact rather than GFR equation (reference 26). Please check throughout the manuscript.
In the methods, reference is made to the important NEJM publication by inker et al (DOI 10.1056/NEJMoa1114248), where the equations are described. A tabular listing is possible. The citations have been checked and (no. 23 and 24 must be 24 and 25; no. 25 must be no. 26) will be corrected.
Line 98 : The expression « the above equations » is confusing because no equation was written, please rephrase.
The following sentence is amended: the equations as described by forde et al.

Reviewer 2 Report
I considered the manuscript entitled “Impact of elevated LDH on Cystatin C-based glomerular filtration rate estimates in patients with cancer” by E. Aydilek, et al that is intended to be published in JCM journal.
The study is simple and clear. Extremely simple. What appears unclear is the importance of the findings. We have been using for decades the eGFRcrea for calculating the renal function, but it was usually criticized. Recently, we have added Cystatin C for this evaluation, which introduces a lot of concerns in some centers since it is difficult to measure in routine area and costly. Now it appears in these patients that the results with Cys are confusing depending on the status of the patients. What is the real and new message? Yet, the under-estimation of renal function with Cystatin may alert the clinicians to use Cystatin C in their patients as worst renal function will bother their decisions. Please clarify
Equations to calculate eGFR must be introduced
Author Response
Response to reviewer 2
The study is simple and clear. Extremely simple. What appears unclear is the importance of the findings. We have been using for decades the eGFRcrea for calculating the renal function, but it was usually criticized. Recently, we have added Cystatin C for this evaluation, which introduces a lot of concerns in some centers since it is difficult to measure in routine area and costly. Now it appears in these patients that the results with Cys are confusing depending on the status of the patients. What is the real and new message? Yet, the under-estimation of renal function with Cystatin may alert the clinicians to use Cystatin C in their patients as worst renal function will bother their decisions. Please clarify
The hopes associated with the introduction of cystatin c-based gfr calculation are dimmed by our results in cancer patients. The study shows - to our knowledge - that in cancer patients with high cell turnover, large tumor mass and associated ldh increase the use of cys-based equation may only be applied with great caution in order to avoid an underestimation of renal function. Underestimation, in turn, is associated with underdosing and consequent worsening of prognosis. Determination of renal function in cancer patients remains difficult and each method (cys or crea based equation or 24-h-colleection urine) has its own difficulties. A workable goldstandard that is widely used does not exist and remains to be defined.
Equations to calculate eGFR must be introduced
In the methods, reference is made to the important NEJM publication by inker et al (DOI 10.1056/NEJMoa1114248), where the equations are described. A tabular listing is possible.
Reviewer 3 Report
In this manuscript, authors examined the results obtained with the different equations and the influence of tumor mass and discuss possible implications for clinical practice. The analysis shows a statistically significant difference in estimates of eGFR by using different eGFR equations in cancer patients. Overall, the content presented in the manuscript could be relevant for clinical implication and considered for publication after minor revision.
Comments to the Authors
1. Page 2, lines 68-70; “serum were collected from all patients as part of routine diagnostics at the time of admission and analyzed in the laboratory of the University Medicine Göttingen. Plasma creatinine, urine creatinine, LDH and cystatin C were analyzed by standard methods”.
Authors need to elaborate the brief analysis methods of Plasma creatinine, urine creatinine, LDH and cystatin C, OR authors have to cite few relevant references.
22. Page 7, 161 and 162; “By using the eGFRcys-based equation there was a significant difference in eGFR between patients with elevated LDH levels compared with patients with normal LDH levels.” Authors need to explain why this significance different in eGFR between patients of normal and abnormal LDH value and how this method would be relevant for the clinical applications.
Author Response
Response to reviewer 3
Page 2, lines 68-70; “serum were collected from all patients as part of routine diagnostics at the time of admission and analyzed in the laboratory of the University Medicine Göttingen. Plasma creatinine, urine creatinine, LDH and cystatin C were analyzed by standard methods”
Authors need to elaborate the brief analysis methods of Plasma creatinine, urine creatinine, LDH and cystatin C, OR authors have to cite few relevant references.
The following sentence is added to the methods: The parameters were measured on the Architect c16000 device from Abbott.
Page 7, 161 and 162; “By using the eGFRcys-based equation there was a significant difference in eGFR between patients with elevated LDH levels compared with patients with normal LDH levels.” Authors need to explain why this significance different in eGFR between patients of normal and abnormal LDH value and how this method would be relevant for the clinical applications
The existing sentence is amended as follows: When using the eGFRcys-based equation, there was a significant difference in eGFR between patients with elevated LDH levels compared with patients with normal LDH levels, caused by an increased level of cystatin C in patients with elevated LDH. This might contribute to the hypothesis that in the presence of increased tumor mass or high cell turnover and associated increased cystatin C, the use of the eGFRcys equation leads to an underestimation of renal function. Thus, the use of the cystatin C-based equation must be used with caution.